# Membrane-partitioned cell wall synthesis in mycobacteria

**Alam García-Heredia[1], Takehiro Kado[2], Caralyn E Sein[2], Julia Puffal[2†], Sarah H Osman[2], Julius Judd[3‡], Todd A Gray[3,4], Yasu S Morita[1,2]\*, M Sloan Siegrist[1,2]\***

[1]Molecular and Cellular Biology Graduate Program, University of Massachusetts, Amherst, United States; [2]Department of Microbiology, University of Massachusetts, Amherst, United States; [3]Division of Genetics, Wadsworth Center, New York State Department of Health, Albany, United States; [4]Department of Biomedical Sciences, University at Albany, Albany, United States

**\*For correspondence:**
ymorita@microbio.umass.edu (YSM);
siegrist@gmail.com (MSS)

**Present address:** †Department of Biochemistry and Molecular Biology, Rutgers University, Robert Wood Johnson Medical School, Piscataway, United States; ‡Department of Molecular Biology and Genetics, Cornell University, Ithaca, United States

**Competing interests:** The authors declare that no competing interests exist.

**Abstract** Many antibiotics target the assembly of cell wall peptidoglycan, an essential, heteropolymeric mesh that encases most bacteria. In rod-shaped bacteria, cell wall elongation is spatially precise yet relies on limited pools of lipid-linked precursors that generate and are attracted to membrane disorder. By tracking enzymes, substrates, and products of peptidoglycan biosynthesis in *Mycobacterium smegmatis*, we show that precursors are made in plasma membrane domains that are laterally and biochemically distinct from sites of cell wall assembly. Membrane partitioning likely contributes to robust, orderly peptidoglycan synthesis, suggesting that these domains help template peptidoglycan synthesis. The cell wall-organizing protein DivIVA and the cell wall itself promote domain homeostasis. These data support a model in which the peptidoglycan polymer feeds back on its membrane template to maintain an environment conducive to directional synthesis. Our findings are applicable to rod-shaped bacteria that are phylogenetically distant from *M. smegmatis*, indicating that horizontal compartmentalization of precursors may be a general feature of bacillary cell wall biogenesis.

## Introduction

The final lipid-linked precursor for peptidoglycan synthesis, lipid II, is made by the glycosyltransferase MurG in the inner leaflet of the plasma membrane. Lipid II is then flipped to the outer leaflet by MurJ where its disaccharide-pentapeptide cargo is inserted into the existing cell wall by membrane-bound transglycosylases and transpeptidases (*Zhao et al., 2017*). Early in vitro work in *Staphylococcus aureus* and *Escherichia coli* indicated that a fluid microenvironment might stimulate the activities of MurG and the upstream, lipid I synthase MraY (*Norris and Manners, 1993*). More recent in vivo data has localized *Bacillus subtilis* MurG to regions of increased fluidity (RIFs, *Müller et al., 2016*; *Strahl et al., 2014*), one of three classes of membrane domains that have been described in bacteria to date. In mycobacteria, intracellular membrane domains (IMD, formerly called PMf, *Morita et al., 2005*) can be separated from the conventional plasma membrane (PM-CW, for plasma membrane associated with cell wall) by sucrose density gradient fractionation. The proteome and lipidome of IMD are distinct from those of the PM-CW (*Hayashi et al., 2016*; *Morita et al., 2005*). Reanalysis of our proteomics data (*Hayashi et al., 2016*) suggested that *Mycobacterium smegmatis* MurG is enriched in the IMD while sequentially acting transglycosylases and transpeptidases associate with the PM-CW. While PM-CW-resident proteins distribute along the perimeter of live mycobacteria, IMD-resident proteins are enriched toward sites of polar cell elongation with additional presence along the sidewalls (*Hayashi et al., 2016*; *Hayashi et al., 2018*). We also noted that the polar enrichment of MurG-RFP resembles that of the validated IMD marker mCherry-GlfT2 or GlfT2-GFP

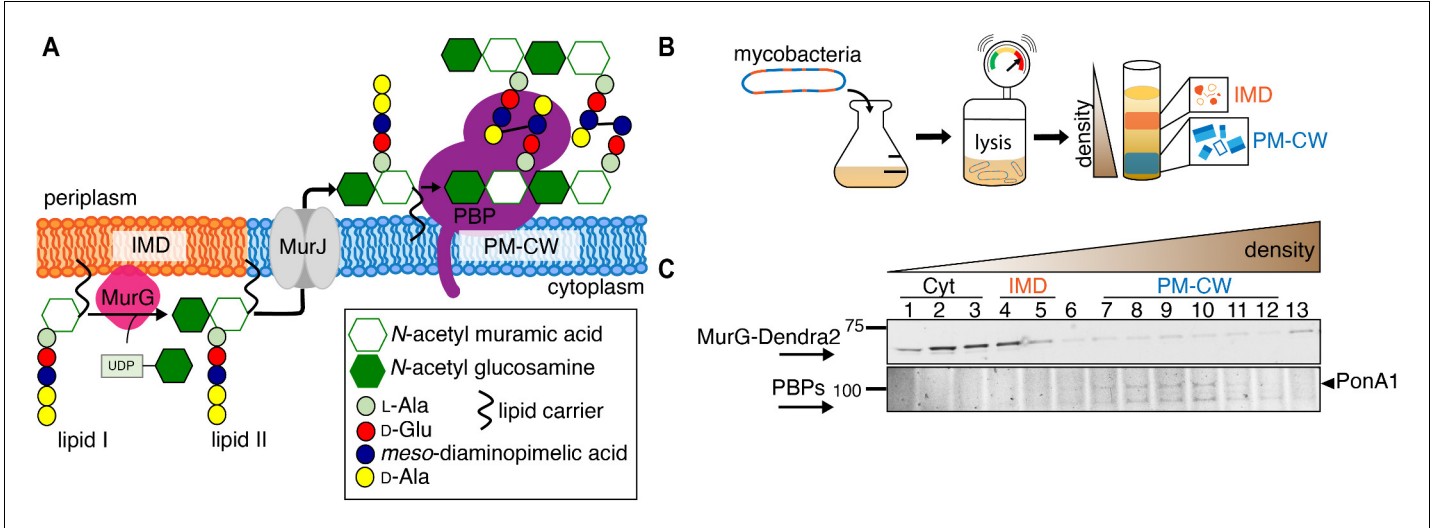

**Figure 1.** MurG is enriched in the IMD, and PBPs associate with PM-CW. (**A**) Membrane-bound steps of peptidoglycan synthesis with hypothesized partitioning into IMD and PM-CW. (**B**) Bacteria are lysed by nitrogen cavitation, and cell lysate is sedimented on a sucrose density gradient. (**C**) Lysates from wild-type or MurG-Dendra2-expressing *M. smegmatis* were fractionated as in (**B**) and separated by SDS-PAGE. Top, in-gel fluorescence shows MurG-Dendra2 association with the IMD. Treatment with benzyl alcohol (BA) redistributed the protein across the fractions. Bottom, wild-type *M. smegmatis* membrane fractions were incubated with Bocillin-FL prior to SDS–PAGE. Labeled PBPs are enriched in PM-CW. Band intensities are quantitated in *Figure 1—figure supplement 4*.

The online version of this article includes the following figure supplement(s) for figure 1:

**Figure supplement 1.** MurG-Dendra2 is functional.

**Figure supplement 2.** Fluorescent fusions do not change the cell length of *M. smegmatis*.

**Figure supplement 3.** Immunoblot analysis of the IMD and the PM-CW membrane fractions separated by sucrose density sedimentation.

**Figure supplement 4.** Membrane-bound MurG-Dendra2 (fractions 4–12) is enriched in the IMD (fractions 4–5), (**A**), and Bocillin-FL-labeled PBPs are enriched in the PM-CW (fractions 7–12), (**B**).

**Figure supplement 5.** MurG-Dendra2 is spatially coincident with the IMD reporter mCherry-GlfT2.

**Figure supplement 6.** Visualization of PonA1 using Bocillin-FL.

(*Hayashi et al., 2016*; *Meniche et al., 2014*), but that nascent peptidoglycan at the mycobacterial poles primarily abuts rather than colocalizes with mCherry-GlfT2 (*Hayashi et al., 2018*). These observations suggested a model where lipid II synthesis is segregated from subsequent steps of cell wall assembly (*Figure 1A*).

## Results and discussion

To test this model, we first expressed a functional MurG-Dendra2 fusion in *M. smegmatis* (*Figure 1—figure supplements 1* and *2*) and assayed its distribution in membrane fractions that had been separated by a sucrose density gradient (*Figure 1B*). MurG-Dendra2, a peripheral membrane protein, was enriched in both the cytoplasmic and IMD membrane fractions (*Figure 1C*, top; *Figure 1—figure supplements 3A* and *4*). In intact cells, polar enrichment of MurG-Dendra2 was coincident with that of the IMD marker mCherry-GlfT2 (*Figure 1—figure supplement 5*). This spatial relationship was similar to that previously observed for other MurG and GlfT2 fluorescent fusion proteins (*Hayashi et al., 2016*; *Hayashi et al., 2018*; *Meniche et al., 2014*).

Enzymes from the penicillin-binding proteins (PBPs) and shape, elongation, division, and sporulation (SEDS) families integrate the disaccharide-pentapeptide from lipid II into peptidoglycan (*Zhao et al., 2017*). While our proteomics did not detect many polytopic membrane proteins, including SEDS proteins, our PM-CW dataset was enriched for all known PBPs (*Hayashi et al., 2016*). Fluorescent derivatives of β-lactam antibiotics, such as Bocillin-FL, bind to PBPs and report transpeptidase-active enzymes. We incubated subcellular fractions from wild-type *M. smegmatis* with Bocillin-FL and identified fluorescent proteins in the PM-CW (*Figure 1C*, bottom; *Figure 1—figure supplements 3B* and *4*). As expected for PBPs, the signal from these bands was diminished by

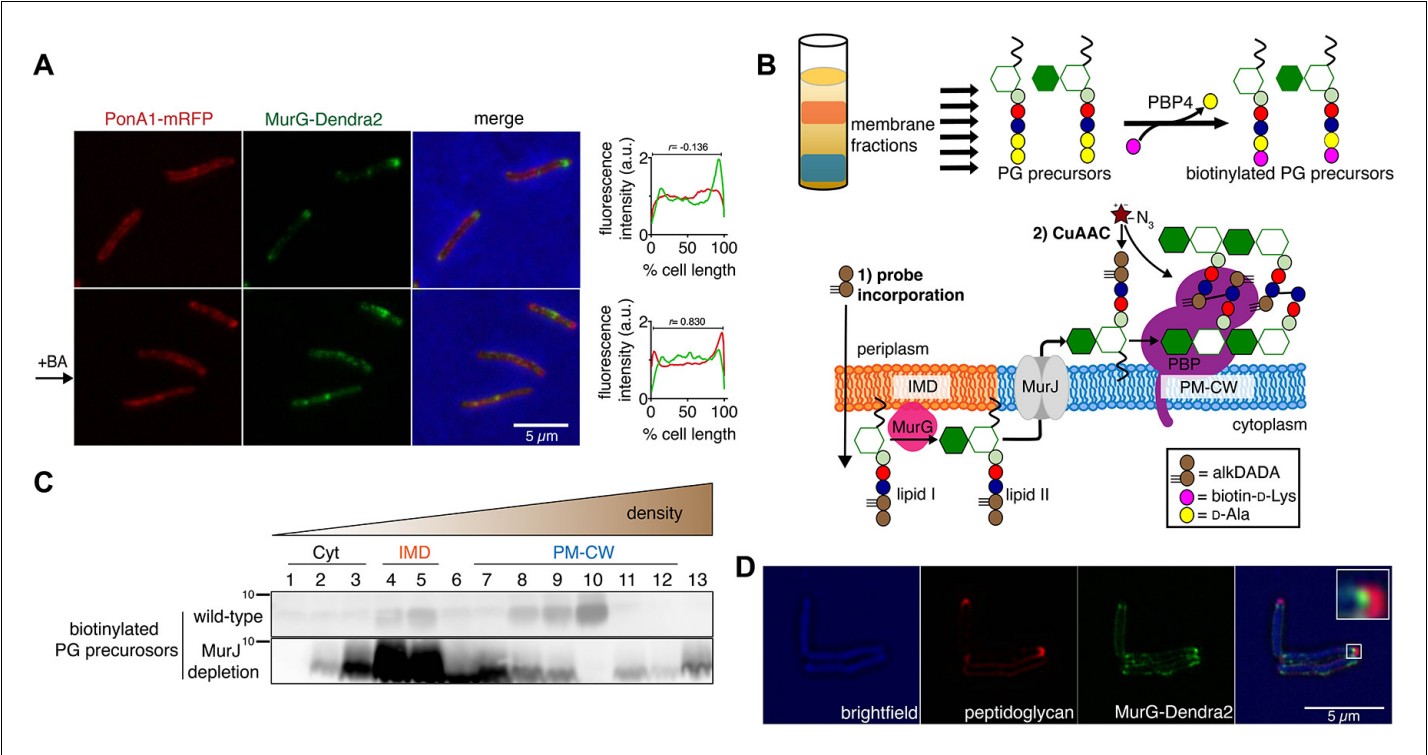

**Figure 2.** Lipid II is synthesized in the IMD and trafficked to the PM-CW. (A) Left, conventional microscopy of *M. smegmatis* coexpressing PonA1-mRFP and MurG-Dendra2 treated +/- benzyl alcohol (BA). Right, fluorescence distribution of the fusion proteins. a.u., arbitrary units. *r* denotes the Pearson's correlation value. 42>n>57. (B) Top, detection of lipid-linked peptidoglycan (PG) precursors from membrane fractions. Bottom, metabolic labeling of mycobacterial cell wall synthesis (*García-Heredia et al., 2018*). (C) PG precursors are labeled as in (B), top. The labeled precursors are in the IMD and PM-CW of wild-type *M. smegmatis* but accumulate in the IMD upon MurJ depletion (*García-Heredia et al., 2018*). While we do not yet understand the loss of signal from fraction 10, we note that there are precursors present but in low abundance (see *Figure 2—figure supplement 3B*). (D) *M. smegmatis*-expressing MurG-Dendra2 were incubated with alkDADA. Surface-exposed alkynes on fixed cells were detected by CuAAC (*García-Heredia et al., 2018*). Bacteria were imaged by SIM-E.

The online version of this article includes the following figure supplement(s) for figure 2:

**Figure supplement 1.** PonA1-mRFP is functional.

**Figure supplement 2.** PimE-GFP is functional and has a similar subcellular localization to PonA1-mRFP.

**Figure supplement 3.** MurJ is critical to comparmentalize both vertical and lateral cell wall synthesis.

pre-treatment with ampicillin (*Figure 1—figure supplement 6*). We focused on characterizing PonA1, an essential bifunctional transglycosylase/transpeptidase in *M. smegmatis* (*Hett et al., 2010*; *Kieser et al., 2015*; *Baranowski et al., 2018*). Depletion of PonA1 (*Hett et al., 2010*) resulted in the loss of the higher molecular band (*Figure 1—figure supplement 6*), confirming this protein is present and active in the PM-CW (*Figure 1C*, bottom). We next expressed a functional PonA1-mRFP fusion in *M. smegmatis* (*Figure 1—figure supplement 2* and *Figure 2—figure supplement 1*; *Kieser et al., 2015*; *Baranowski et al., 2018*). Although we detected potential breakdown products of the fusion protein by anti-RFP immunoblot (*Figure 2—figure supplement 1*), we found that it, like native PonA1, was active in the PM-CW and distributed along the sidewall in a manner similar to the functional PM-CW marker PimE-GFP (*Figure 2—figure supplements 1* and *2*, *Hayashi et al., 2016*). Coexpression of MurG-Dendra2 and PonA1-mRFP confirmed that the proteins have different subcellular localization (*Figure 2A*). Together, our data show that MurG and PonA1 occupy membrane compartments that are biochemically and likely spatially distinct.

The association of MurG with the IMD and of PonA1 with the PM-CW implies that the IMD is the site of lipid II synthesis, while the PM-CW is where peptidoglycan assembly takes place. We refined an in vitro D-amino acid exchange assay to detect lipid-linked peptidoglycan precursors from membrane fractions (*Figure 2B*, *García-Heredia et al., 2018*; *Qiao et al., 2014*). In wild-type cells, we detected biotinylated molecules in both the IMD and PM-CW (*Figure 2C*; *Figure 1—figure*

supplement 3B). We hypothesized that the labeled species comprise precursors in both the inner and outer leaflets of the plasma membrane. We and others have shown that depletion of MurJ results in accumulation of biotinylated precursors (*García-Heredia et al., 2018*; *Qiao et al., 2017*). By performing the D-amino acid exchange reaction on membrane fractions obtained from MurJ-depleted *M. smegmatis* (*Figure 2—figure supplement 3A*), we found that precursors accumulate in the IMD (*Figure 2C*; *Figure 1—figure supplement 3C*; *Figure 2—figure supplement 3B*). These results suggest that lipid II is made in the IMD and transferred to the PM-CW in a MurJ-dependent manner.

Based on our biochemical data, we hypothesized that lipid II incorporation into the cell wall is laterally segregated from its synthesis. We previously showed that alkynyl and azido D-amino acid dipeptides (*Liechti et al., 2014*) incorporate into lipid-linked peptidoglycan precursors in *M. smegmatis* (*García-Heredia et al., 2018*) and that metabolic labeling with alkynyl dipeptide (alkDADA or EDA-DA, *Liechti et al., 2014*) is most intense in regions adjacent to the IMD marker mCherry-GlfT2 (*Hayashi et al., 2018*). We labeled MurG-Dendra2-expressing *M. smegmatis* with alkDADA and detected the presence of the alkyne by copper-catalyzed azide-alkyne cycloaddition (CuAAC, *García-Heredia et al., 2018*). To distinguish extracellular alkynes present in periplasmic lipid II and newly polymerized cell wall from alkynes originating from cytoplasmic lipid II, we selected picolyl azide-Cy3 as our label because of its poor membrane permeability (*Figure 2B*, *Yang and Hinner, 2015*). Using this optimized protocol, we observed nascent peptidoglycan deposition at the polar tip, whereas MurG-Dendra2 was proximal to this site (*Figure 2D*). Our data suggest that lipid II synthesis is laterally partitioned from the subsequent steps of peptidoglycan assembly. MurJ depletion reduced and delocalized alkDADA-derived fluorescence (*Figure 2—figure supplement 3C*), consistent with a gatekeeper role for the flippase in both lateral membrane compartmentalization and flipping across the inner membrane.

Next, we wanted to understand the significance of membrane architecture for cell wall synthesis. We perturbed the membrane with benzyl alcohol, a compound that preferentially inserts into disordered membrane regions in vitro (*Muddana et al., 2012*) and has been used to fluidize membranes from mammalian and bacterial cells (*Friedlander et al., 1987*; *Ingram, 1976*; *Konopásek et al., 2000*; *Nagy et al., 2007*; *Strahl et al., 2014*; *Zielińska et al., 2020*). In *B. subtilis*, benzyl alcohol disrupts RIFs (*Müller et al., 2016*). In *M. smegmatis*, we found that benzyl alcohol reduced the cellular material associated with the IMD (*Figure 3A*, *Figure 3—figure supplement 1*, *Figure 1—figure supplement 3D*) and altered the distribution of FM4-64, a non-specific lipophilic dye (*Figure 3—figure supplement 2*), and of plasma membrane glycolipids (*Figure 3—figure supplement 3*). However, the fluidizer did not alter labeling by N-AlkTMM or O-AlkTMM (*Figure 3—figure supplement 4*), probes that, respectively, mark the noncovalent and covalent lipids of the outer 'myco' membrane (*Foley et al., 2016*). These observations suggest that benzyl alcohol primarily affects the plasma membrane in *M. smegmatis*. MurG-Dendra2 was also less enriched in the IMD fraction following benzyl alcohol treatment (*Figure 1C*, top; *Figure 1—figure supplements 3E* and *4*) and, in live cells, at the poles (*Figure 2A*). By contrast, benzyl alcohol produced subtle changes in the subcellular distribution of active PBPs (*Figure 1C*, bottom; *Figure 1—figure supplements 3D* and *4*), although PonA1 shifted toward the poles in live cells (*Figure 2A*). Disruption of plasma membrane architecture was accompanied by dampening and delocalization of peptidoglycan assembly (*Figure 3B*, *Figure 3—figure supplement 5*) as well as a reduction in lipid precursor synthesis (*Figure 3C*) and halt in polar elongation (*Figure 3—figure supplement 4*). The effects of benzyl alcohol were reversible, as indicated by colony-forming units and prompt recovery of peptidoglycan synthesis (*Figure 3—figure supplement 6*). Dibucaine, a compound that preferentially disrupts ordered membrane regions in vitro (*Kinoshita et al., 2019*) and can fluidize membranes from eukaryotic cells (*Kim et al., 1997*), also delocalized IMD-resident proteins (*Figure 3—figure supplement 7*) and delocalized and reduced peptidoglycan synthesis (*Figure 3B*, *Figure 3—figure supplement 5*).

Our data suggest that membrane architecture contributes to peptidoglycan synthesis and cell growth in *M. smegmatis*. While we cannot rule out pleiotropic effects of chemical fluidizers on membrane potential or membrane protein activity, we note that benzyl alcohol decreases peptidoglycan precursor accumulation (*Figure 3C*), rather than increasing it as occurs with protonophore treatment (*Rubino et al., 2018*) or MurJ depletion (*Figure 2C*; *García-Heredia et al., 2018*; *Qiao et al., 2017*); MurG and PonA1 are retained in the membrane upon benzyl alcohol treatment (*Figure 1C*,

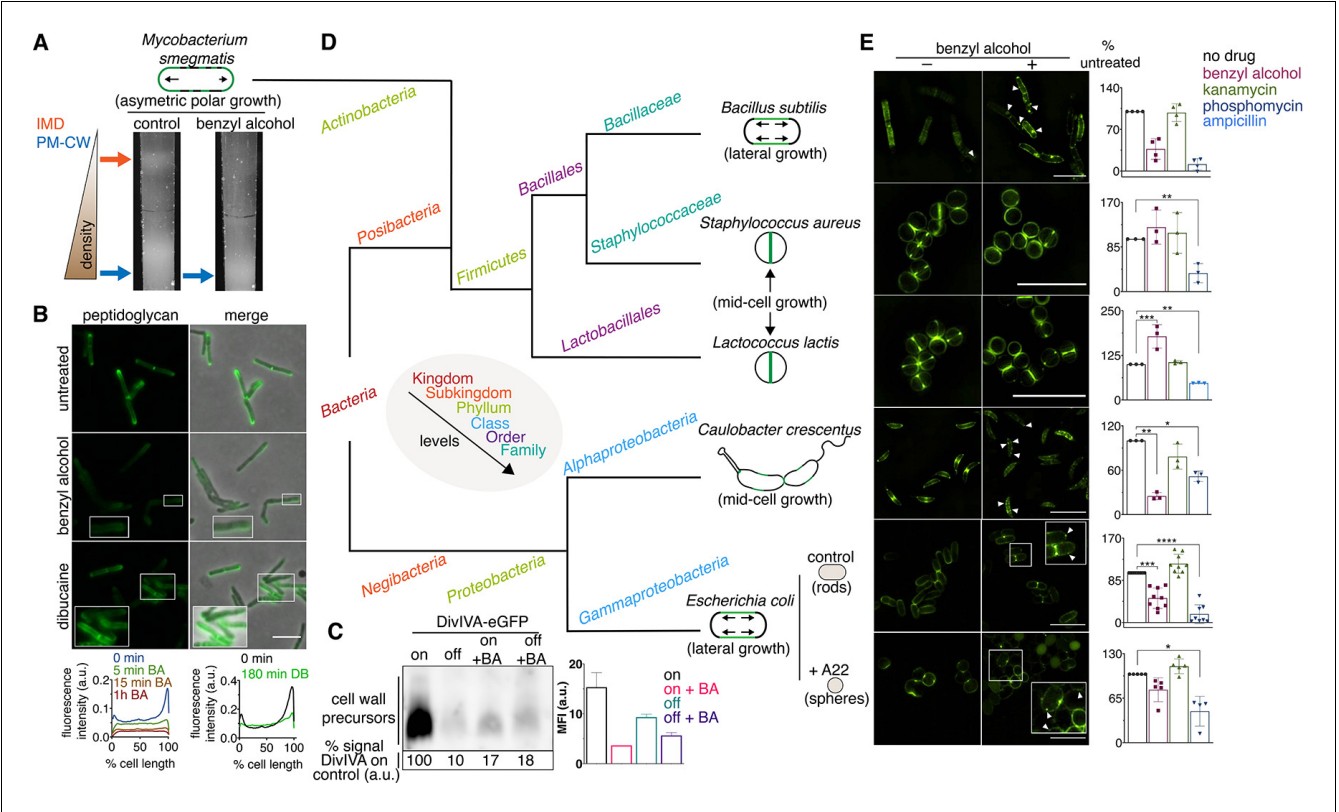

**Figure 3.** Membrane perturbations disrupt peptidoglycan biogenesis in *M. smegmatis* and phylogenetically-distant bacilli. (A) Lysates from wild-type *M. smegmatis* treated +/- benzyl alcohol (BA) were sedimented in a sucrose density gradient. Density of the cellular material is quantified in *Figure 3—figure supplement 1*. (B) Top, wild-type *M. smegmatis* was incubated or not with benzyl alcohol or dibucaine, then labeled with alkDADA; merged images correspond to fluorescent image with the corresponding phase contrast. Bottom, the distribution of peptidoglycan labeling from wild-type *M. smegmatis* that was incubated with BA or dibucaine (DB) for the indicated time was quantitated as in *Figure 2A*, except that signal intensity was not normalized. The changes in fluorescence are further quantified by flow cytometry in *Figure 3—figure supplement 5*. (C) Top left, DivIVA-eGFP-ID *M. smegmatis* was either treated with benzyl alcohol, depleted of DivIVA, or both, and the peptidoglycan precursors from whole cells were biotinylated as in *Figure 2C*. Bottom left, biotin-derived chemiluminescence was quantified by densitometry; signal is expressed as % of untreated DivIVA-eGFP-ID (first lane). Right, DivIVA-eGFP-ID *M. smegmatis* was treated as in the left panel but labeled with alkDADA, subjected to CuAAC, and analyzed by flow cytometry. MFI, median fluorescence intensity values for a representative experiment. Error bars denote standard deviation of technical triplicates. (D) Phylogenetic tree constructed with 16S rDNA sequences (rate of mutation not considered). Taxonomic groups matched with colors to their levels with only diverging points shown. Shapes and growth modes illustrated for select species. (E) Left, different bacteria treated +/- benzyl alcohol followed by alkDADA incubation. Arrowheads highlight irregular patches of peptidoglycan. Insets are magnified. Where applicable, *E. coli* was pre-incubated with A22. Right, bacteria were treated with benzyl alcohol, translation-inhibiting kanamycin, or peptidoglycan-acting phosphomycin or ampicillin and then labeled as in (B) and analyzed by flow cytometry (see Materials and methods). MFI values were normalized to untreated controls. Experiments were performed three to nine times in triplicate. Error bars denote standard deviation of biological replicates. *p<0.05; **p<0.005; ***p<0.0005; ****p<0.00005, ratio paired t-tests and one-way ANOVA with Dunnet's test for non-normalized MFI of biological replicates. Scale bars, 5 μm.

The online version of this article includes the following figure supplement(s) for figure 3:

**Figure supplement 1.** The effects of benzyl alcohol, DivIVA depletion, and spheroplasting on IMD and PM-CW abundance.

**Figure supplement 2.** Benzyl alcohol alters FM4-64FX distribution.

**Figure supplement 3.** Benzyl alcohol and depletion of DivIVA affect the distribution of membrane glycolipids.

**Figure supplement 4.** Benzyl alcohol halts cell elongation but does not otherwise impact the localization of mycomembrane probes.

**Figure supplement 5.** Benzyl alcohol and dibucaine decrease peptidoglycan synthesis over time.

**Figure supplement 6.** *M. smegmatis* survives benzyl alcohol (BA) treatment.

**Figure supplement 7.** Polar enrichment and spatial coincidence of MurG-Dendra2 and mCherry-GlfT2 decrease upon dibucaine treatment.

**Figure supplement 8.** Benzyl alcohol does not delocalize DivIVA-eGFP.

*Figure 1—figure supplement 4*); and at least a subset of membrane-bound PBPs remain competent for Bocillin-FL binding (*Figure 1C*, *Figure 1—figure supplement 4*). Moreover, benzyl alcohol and dibucaine delocalize nascent peptidoglycan from the poles to the sidewall (*Figure 3B*), an effect that

cannot be explained by diminished synthesis alone. In the case of benzyl alcohol, redistribution of new cell wall occurs within minutes (*Figure 3B*, *Figure 3—figure supplement 5*), consistent with rapid fluidization (*Konopásek et al., 2000*; *Yano et al., 2016*), and likely prior to changes in gene expression. Nevertheless, it is possible that chemical fluidizers directly disrupt the activity of cell wall synthesis proteins in addition to altering the milieu in which these proteins function.

The impacts of benzyl alcohol and dibucaine on MurG-Dendra2 localization were subtly different (compare *Figure 2A* to *Figure 3—figure supplement 7*) as was the time frame for disruption of cell wall synthesis by these chemicals (*Figure 3B*, *Figure 3—figure supplement 5*). In model membranes, benzyl alcohol promotes phase separation by further fluidizing disordered regions (*Muddana et al., 2012*), while dibucaine disrupts phase separation by fluidizing ordered regions (*Kinoshita et al., 2019*). In more-complex cellular membranes, the effects of these compounds may be influenced by the presence of pre-existing mechanisms that establish and maintain membrane organization. For example, the architecture of eukaryotic membranes is influenced by transient links, or pinning, to the cytoskeleton (*Fujimoto and Parmryd, 2016*; *Liu et al., 2015*). In *B. subtilis* and *E. coli*, actin homologs like MreB direct peptidoglycan synthesis along the lateral cell surface (*Daniel and Errington, 2003*; *Iwai et al., 2002*; *Shi et al., 2018*; *Zhao et al., 2017*). They also organize the membrane into domains of increased (*Oswald et al., 2016*; *Strahl et al., 2014*) and decreased fluidity (*Wagner et al., 2020*). Global reductions in membrane fluidity interfere with the assembly and motion of *B. subtilis* MreB (*Zielińska et al., 2020*; *Gohrbandt et al., 2019*; *Kurita et al., 2020*), potentially indicating a feedback loop between the physical state of the membrane and MreB-directed cell wall elongation. We found that benzyl alcohol delocalized or dampened cell wall assembly in rod-shaped bacteria with divergent envelope composition and modes of growth (*Figure 3D,E*). Peptidoglycan synthesis was less affected by benzyl alcohol in coccoid species, which lack MreB or obvious RIFs (*Wenzel et al., 2018*), and in rounded, A22-treated *E. coli*, in which MreB assembly is inhibited (*Figure 3E*). Thus, membrane organization likely contributes to effective, directional cell wall synthesis in rod-shaped bacteria.

Mycobacteria lack MreB. How, then, is the IMD partitioned away from the rest of the plasma membrane? In these organisms, the essential tropomyosin-like protein DivIVA (Wag31) concentrates cell wall assembly at the poles (*Jani et al., 2010*; *Kang et al., 2008*; *Melzer et al., 2018*). DivIVA depletion results in deformation and rounding of mycobacterial cells (*Kang et al., 2008*; *Meniche et al., 2014*; *Nguyen et al., 2007*). Given the similarities in DivIVA and MreB function, we hypothesized that DivIVA creates and/or maintains the IMD. We used *M. smegmatis* expressing DivIVA-eGFP-ID in which DivIVA is fused to both eGFP and an inducible degradation tag (*Meniche et al., 2014*) to deplete DivIVA. Depletion of the protein reduced the amount of IMD-associated cellular material (*Figure 4A*, *Figure 3—figure supplement 1*, *Figure 1—figure supplement 3F–G*), altered the distribution of plasma membrane glycolipids (*Figure 3—figure supplement 3*), and delocalized the IMD marker mCherry-GlfT2 from the poles (*Figure 4B*).

DivIVA phosphorylation regulates MraY and/or MurG activity via an indirect, unknown mechanism (*Jani et al., 2010*). Consistent with these data, we found that depletion of the protein reduced both lipid-linked peptidoglycan precursor abundance and alkDADA incorporation (*Figure 3C*). Membrane disruption by benzyl alcohol did not delocalize DivIVA from the polar tips (*Figure 3—figure supplement 8*), and the suppressive effects of benzyl alcohol and DivIVA depletion on precursor abundance and cell wall synthesis were not additive (*Figure 3C*), suggesting that the perturbations act on the same pathway. Unlike DivIVA depletion (*Figure 4B*), benzyl alcohol does not change *M. smegmatis* shape (*Figure 2A*). Therefore, while we cannot exclude the possibility that spherical morphology in DivIVA-depleted cells indirectly impacts membrane partitioning – for example by mislocalization of curvature-sensing proteins or by altering the spacing between the membrane and cell wall – our results are most consistent with a model in which DivIVA organizes the mycobacterial membrane for optimal cell wall synthesis.

As lipid II both generates and homes to disordered regions of model membranes (*Ganchev et al., 2006*; *Jia et al., 2011*; *Valtersson et al., 1985*), the effect of DivIVA on precursors suggests that concentrated peptidoglycan synthesis is a cause or a consequence (or both) of IMD/PM-CW partitioning. In other organisms, lipid II production is required for MreB rotation (*Domínguez-Escobar et al., 2011*; *Garner et al., 2011*; *van Teeffelen et al., 2011*), to recruit MreB to the plasma membrane (*Schirner et al., 2015*), and for normal membrane staining by a lipophilic fluorescent dye (*Muchová et al., 2011*), so the precursor might also play an indirect role in

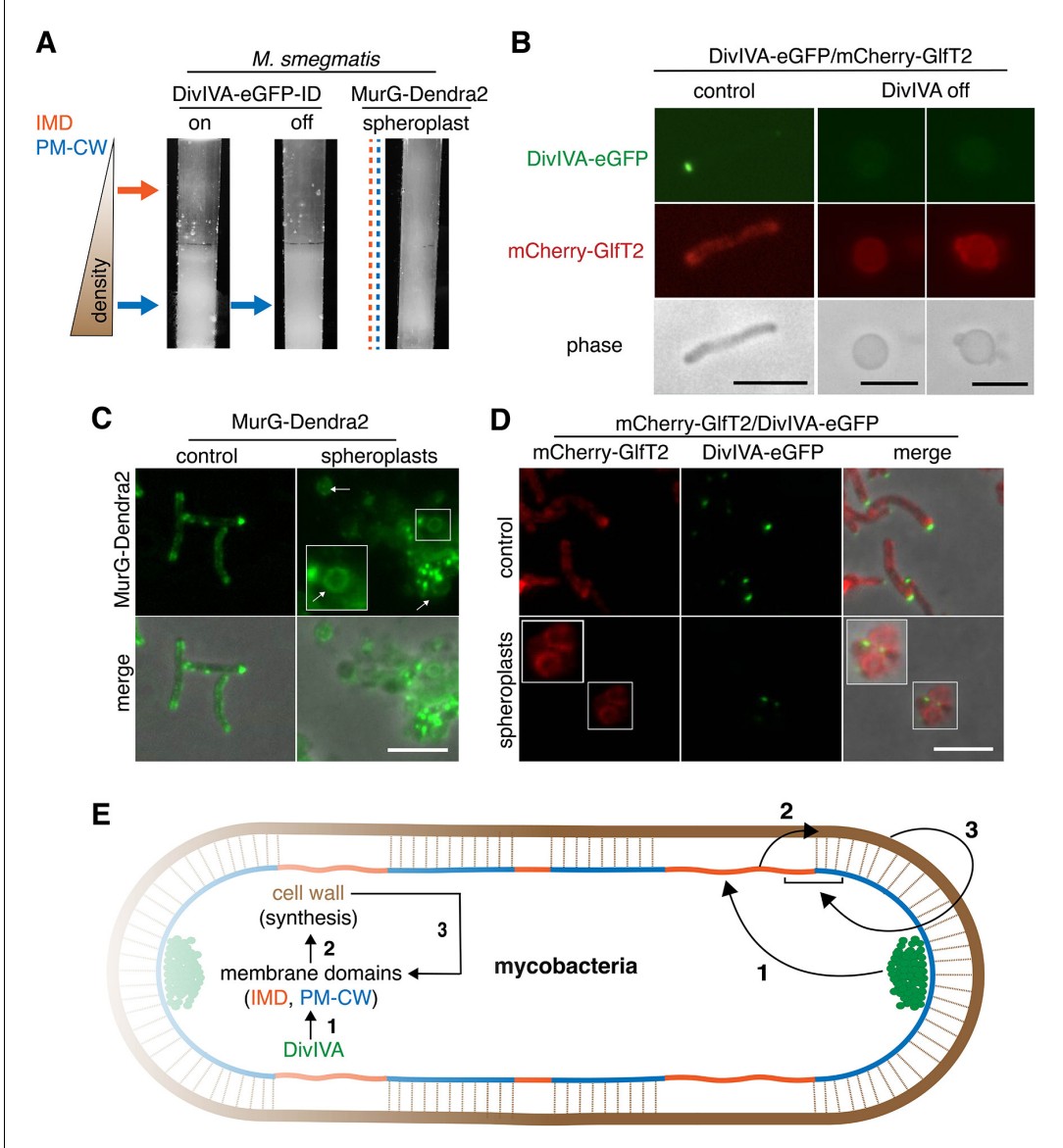

**Figure 4.** DivIVA and an intact cell wall promote membrane domain maintenance. (**A**) Lysates from MurG-Dendra2-expressing *M. smegmatis* spheroplasts (*Melzer et al., 2018*) or from the DivIVA-eGFP-ID strain depleted (off) or not (on) of DivIVA (*Meniche et al., 2014*) were sedimented in a sucrose density gradient. (**B**) DivIVA was depleted or not from mCherry-GlfT2-expressing *M. smegmatis*. Depletion of DivIVA delocalizes mCherry-GlfT2. *M. smegmatis* expressing MurG-Dendra2 (**C**) or coexpressing mCherry-GlfT2 and DivIVA-eGFP-ID (**D**) were spheroplasted or not (control) and imaged. In spheroplasted cells, the IMD-associated proteins distribute along the cell periphery. Arrows mark spheroplasts outside and within insets, which have increased size and brightness. Merged images correspond to fluorescent image with the corresponding phase contrast. (**E**) Model for self-organization of plasma membrane and cell wall in *M. smegmatis*. Brown line indicates the cell wall. Short brown lines perpendicular to the membrane and cell wall indicate that the cell wall is likely to be physically connected to the membrane in the PM-CW regions (*Morita et al., 2005*). All scale bars, 5 μm. The online version of this article includes the following figure supplement(s) for figure 4:

**Figure supplement 1.** An intact cell wall supports MurG partitioning within the membrane.

compartmentalizing the mycobacterial membrane via its influence on DivIVA. However, we previously found that the IMD is biochemically intact after 8 hr of treatment with D-cycloserine (*Hayashi et al., 2018*), an antibiotic that we have shown to block *M. smegmatis* peptidoglycan precursor synthesis within 1 hr (*García-Heredia et al., 2018*). IMD-resident proteins delocalize, but not until 6 hr of treatment. The persistence of IMD-resident proteins and the time frame of delocalization

indicate that concentrated lipid II synthesis is more likely a consequence, rather than a cause, of mycobacterial membrane compartmentalization.

Our data supported a model in which DivIVA is necessary to maintain membrane partitioning, which in turn supports efficient synthesis of peptidoglycan precursors and their precise incorporation into the cell wall. However, we noted that eukaryotic membrane architecture can also be influenced by transient interactions with external structures like extracellular matrix and cellulose (*Fujimoto and Parmryd, 2016*; *Liu et al., 2015*). Furthermore, biophysical modeling suggests that osmotic pinning of the plasma membrane against the bacterial cell wall can induce microphase separation (*Mukhopadhyay et al., 2008*). In mycobacteria, co-fractionation of the plasma membrane and cell wall (i.e. PM-CW) upon mechanical cell lysis implies that they are physically connected (*Morita et al., 2005*). We wondered whether the peptidoglycan polymer itself might contribute to the membrane partitioning that organizes its synthesis. In *B. subtilis*, for example, enzymatic removal of the cell wall delocalizes membrane staining by a lipophilic fluorescent dye (*Muchová et al., 2011*) and enhances the mobility of membrane domain-associated flotillin proteins (*Wagner et al., 2020*). To test this hypothesis in *M. smegmatis*, we spheroplasted bacteria that expressed MurG-Dendra2 or DivIVA-eGFP-ID and mCherry-GlfT2. All fusions were functional (*Figure 1—figure supplement 1*, *Hayashi et al., 2016*; *Meniche et al., 2014*). Fractionated lysates from spheroplasted mycobacteria had more diffuse distribution of cellular material and indistinct separation of the IMD and PM-CW fractions (*Figure 4A*, *Figure 3—figure supplement 1*). Consistent with the macroscopic appearance of the fractionated lysate, MurG-Dendra2 was distributed throughout the gradient (*Figure 4—figure supplement 1*; *Figure 1—figure supplement 3H*). MurG-Dendra2 and the IMD marker mCherry-GlfT2 were also diffusely distributed around the periphery of spheroplasted cells while DivIVA, likely a PM-CW-associated protein (*Hayashi et al., 2016*), remained in foci (*Figure 4C,D*). These experiments suggest that an intact cell wall and DivIVA promote membrane compartmentalization in *M. smegmatis* (*Figure 4E*, arrows 1 and 3).

While peptidoglycan biogenesis is well known to vertically span the inner and outer leaflets of the plasma membrane, here we demonstrate in *M. smegmatis* that it is also horizontally partitioned (*Figure 4E*, arrow 2). Partitioning of the mycobacterial membrane by DivIVA and the cell wall follows similar logic to that of eukaryotic membranes, which can be compartmentalized by pinning to cytoplasmic structures such as the actin cytoskeleton and to external structures such as the extracellular matrix and cellulose (*Fujimoto and Parmryd, 2016*; *Liu et al., 2015*), and of model lipid bilayers, which can be phase separated by adhesive forces (*Gordon et al., 2008*; *Mukhopadhyay et al., 2008*). In mycobacteria, the membrane regions that promote cell wall synthesis are likely segregated by the end product of the pathway (*Figure 4E*, arrow 3). The conservation of the cell wall synthesis machinery and elongation-promoting cytoskeletal proteins among phylogenetically distant species predicts that our findings will generally apply to bacilli beyond our mycobacterial model. For rod-shaped species, our model is that the membrane-cell wall axis is a self-organizing system in which directed cell wall synthesis organizes the plasma membrane, and an organized plasma membrane in turn makes cell wall elongation more efficient and precise.

## Materials and methods

**Key resources table**

| Reagent type (species) or resource | Designation | Source or reference | Identifiers | Additional information |
|---|---|---|---|---|
| Strain (*M. smegmatis* mc²155) | *M. smegmatis* | NC_008596 in GenBank | | Wild-type *M. smegmatis* |
| Strain (*M. smegmatis*) | MurG-Dendra2 | This study | | The mutant was generated as described in Supplementary material and methods. |
| Strain (*M. smegmatis*) | mCherry-GlfT2 | *Hayashi et al., 2016* | | See reference for details. |

*Continued on next page*

*Continued*

| Reagent type (species) or resource | Designation | Source or reference | Identifiers | Additional information |
|---|---|---|---|---|
| Strain (*M. smegmatis*) | PonA1-mRFP | *Kieser et al., 2015*; *Baranowski et al., 2018* | | Obtained from Dr. Eric Rubin (Harvard SPH) and Dr. Hesper Rego (Yale Med). |
| Strain (*M. smegmatis*) | PimE-GFP | This study | | The strain was generated as described in Supplementary material and methods. |
| Strain (*M. smegmatis*) | MurG-ID depletion strain | *Meniche et al., 2014* | | Obtained from Dr. Chris Sassetti (U Mass Med) |
| Strain (*M. smegmatis*) | MurJ-ID (MviN) depletion strain | *Gee et al., 2012* | | Obtained from Dr. Chris Sassetti (U Mass Med) |
| Strain (*M. smegmatis*) | p$_{tet}$ponA1 | *Hett et al., 2010* | | Obtained from Dr. Eric Rubin (Harvard SPH) |
| Strain (*M. smegmatis*) | DivIVA-eGFP-ID | *Meniche et al., 2014* | | Obtained from Dr. Chris Sassetti (U Mass Med) |
| Strain (*M. smegmatis*) | mCherry-GlfT2/DivIVA-eGFP-ID | This study | | See reference for details. |
| Strain (*B. subtilis* JH642) | *B. subtilis* | NZ_CP007800 in GeneBank | | |
| Strain (*C. crescentus*) | *C. crescentus* | NA 1000 | | Obtained from Dr. Peter Chien (U Mass Amherst) |
| Strain (*E. coli* K12) | *E. coli* K12 | MG1655 | | |
| Strain (*S. aureus*) | *S. aureus* | ATCC BA-1718 | | Obtained from Dr. Thai Thayumanavan (U Mass Amherst) |
| Strain (*L. lactis*) | *L. lactis lactis* | NRRL B633 | | |
| Chemical compound | Alkyne-D-alanine-D-alanine (alkDADA or EDA-DA) | *Liechti et al., 2014* | | Synthesized by the Chemical Synthesis Core Facility at Albert Einstein College of Medicine (The Bronx, NY) following the referenced protocols. |
| Chemical compound | O-alkyne-trehalose monomycolate (O-AlkTMM) | *Foley et al., 2016* | | Obtained from Dr. Benjamin Swarts (Central Michigan University). |
| Chemical compound | N-alkyne-trehalose monomycolate (N-AlkTMM) | *Foley et al., 2016* | | Obtained from Dr. Benjamin Swarts (Central Michigan University). |
| Software, algorithm | MATLAB codes | *García-Heredia et al., 2018* | | Scripts designed for MATLAB to analyze the fluorescence profiles along a cell body from data collected in Oufti (*Paintdakhi et al., 2016*). |
| Chemical compound | Fmoc-D-Lys(biotinyl)-OH BDL precursor | Chem-Impex International (Wood Dale, IL) | Cat # 16192 | Deprotected as described in *Qiao et al., 2014* to yield BDL. |
| Chemical compound | A22 (S-3,4-Dichlorobenzylisothiourea) | Sigma-Aldrich, St. Louis, MO | SML0471 | Dissolved in water and kept at −20℃. |
| Recombinant DNA reagent | PBP4 plasmid | *Qiao et al., 2014* | | Obtained from Dr. Suzanne Walker (Harvard Med). |

## Bacterial strains and growth conditions

*Mycobacterium smegmatis* mc²155 was grown in Middlebrook 7H9 growth medium (BD Difco, Franklin Lakes, NJ) supplemented with 0.4% (vol/vol) glycerol, 0.05% (vol/vol) Tween-80 (Sigma–Aldrich, St. Louis, MO), and 10% albumin-dextrose-catalase, as well as apramycin (50 μg/mL), kanamycin (25 μg/mL; Sigma–Aldrich, St. Louis, MO), and hygromycin (50 μg/mL) where appropriate. *Staphylococcus aureus* ATCC BA-1718 was grown in BHI (BD Difco, Franklyn Lakes, NJ); *Escherichia coli* K12 and *Bacillus subtilis* ZB307 in LB (VWR, Radnor, PA); *Caulobacter crescentus* NA 1000 in peptone yeast extract (BD Difco, Franklyn Lakes, NJ); and *Lactococcus lactis* NRRL B633 in MRS (Oxoid, Basingstoke, Hampshire, UK). All bacteria were grown shaking at 37°C with the exception of *C. crescentus*, which was incubated at 30°C. See Key Resources Table.

## Mutant strain construction

To test the function of PimE-GFP-FLAG fusion, an expression vector for PimE-GFP-FLAG (pYAB186, *Hayashi et al., 2016*) was electroporated into a *pimE* deletion mutant (*Morita et al., 2006*). Three independent colonies were picked for the phenotypic complementation of AcPIM6 biosynthetic defects. Lipid purification and analysis were performed as described previously (*Morita et al., 2006*).

The *murG* gene was amplified by PCR from *M. smegmatis* mc²155 genomic DNA by PCR, excluding the stop codon, and was inserted into pMSR in-frame with mycobacterial codon-optimized Dendra2 using In-Fusion cloning (Takara Bio, Mountain View, CA). This construct was transformed by electroporation into *M. smegmatis* mc²155, where it integrates at the L5 *attB* site, and was selected by apramycin treatment. Constitutive expression of the MurG-Dendra2 fusion was achieved through the Psmyc promoter (GenBank: AF395207.1). The plasmid construct was validated by Sanger sequencing.

To replace the endogenous *glfT2* gene with a gene-encoding HA-mCherry-GlfT2 in the DivIVA-eGFP-ID strain, we electroporated pMUM052 (*Hayashi et al., 2016*) into DivIVA-eGFP-ID *M. smegmatis*, and positive clones were isolated using hygromycin resistance marker and SacB-dependent sucrose sensitivity. Correct replacement of the *glfT2* gene was confirmed by PCR.

## Generation of spheroplasts

To generate spheroplasts, we followed a previous protocol (*Melzer et al., 2018*). Briefly, wild-type, MurG-Dendra2-expressing, or mCherry-GlfT2/DivIVA-eGFP-ID-coexpressing *M. smegmatis* was grown until log phase. Glycine (1.2% wt/vol final concentration) was added, and the culture was incubated for 24 hr at 37°C with shaking. Afterwards, the cells were washed with sucrose–MgCl₂–maleic acid (SMM) buffer (pH 6.8) and harvested by centrifugation (4000 x *g* for 5 min). The pellet was resuspended in 7H9 medium where the water was replaced with SMM buffer; the medium also was supplemented with glycine (1.2% final concentration) and lysozyme (50 μg/mL final concentration). Bacteria were incubated another 24 hr at 37°C with shaking, and then spheroplasts were either imaged by conventional fluorescence microscopy or lysed by nitrogen cavitation immediately for subsequent biochemical analysis.

## Membrane fractionation

Log-phase *M. smegmatis* (that, where applicable, were untreated, treated with benzyl alcohol, depleted for DivIVA or spheroplasted; see *Supplementary file 1a*) were harvested by centrifugation and washed in phosphate-buffered saline (PBS) + 0.05% Tween-80 (PBST). One gram of wet pellet was resuspended in 5 mL of lysis buffer containing 25 mM HEPES (pH 7.4), 20% (wt/vol) sucrose, 2 mM EGTA, and a protease inhibitor cocktail (ThermoFisher Scientific, Waltham, MA) as described (*Morita et al., 2005*). Bacteria were lysed using nitrogen cavitation at 2000 psi for 30 min three times. Cell lysates were centrifuged at 3220 x *g* for 10 min at 4°C twice to remove unlysed cells prior to loading on a 20–50% sucrose gradient. Membrane-containing fractions were collected in 1 mL after ultracentrifugation at 35,000 rpm on an SW-40 rotor (Beckman, Brea, CA) for 6 hr at 4°C and stored at −80°C prior to analysis.

## Detection of proteins in membrane fractions

MurG-Dendra2 and penicillin-binding proteins (PBPs) were detected by in-gel fluorescence. For MurG-Dendra2, membrane fractions were incubated with an equal volume of 2× loading buffer and then separated by SDS–PAGE on a 12% polyacrylamide gel. To detect PBPs, 50 µg total protein from wild-type *M. smegmatis* membrane fractions were incubated with 40 µM of Bocillin-FL (ThermoFisher Scientific, Waltham, MA) for 30 min in the dark at 37°C. An equal volume of 2× loading buffer was then added, and the mixture was boiled for 3 min at 95°C and then incubated on ice for 30 min. Membrane mixtures were separated on a 12% polyacrylamide gel. Gels were washed in distilled water and imaged using an ImageQuant LAS 4000mini (GE Healthcare, Chicago, IL).

MurJ, PimB′, and MptA were detected by immunoblot. Briefly, cell lysate or membrane fraction proteins were separated by SDS–PAGE on a 12% polyacrylamide gel and transferred to a PVDF membrane. The membrane was blocked with 3% milk in PBS + 0.05% Tween-80 (PBST) and then incubated overnight with primary antibodies (monoclonal mouse anti-FLAG [to detect MurJ; Sigma], and polyclonal rabbit anti-PimB′ or anti-MptA antibodies) (*Sena et al., 2010*). Antibodies were detected with appropriate secondary antibodies conjugated to horseradish peroxidase (GE Healthcare, Chicago, IL). Membranes were rinsed in PBS + 0.05% Tween-20 and visualized by ECL in an ImageQuant LAS 4000mini (GE Healthcare, Chicago, IL) as above.

## Cell envelope labeling

Chemical probes used in this work include alkDADA (EDA-DA [*Liechti et al., 2014*], N-AlkTMM and O-AlkTMM [*Foley et al., 2016*], and FM4-64 FX [Invitrogen, Carlsbad, CA]). AlkDADA was synthesized by the Einstein Chemical Biology Core, and the TMM probes were kind gifts of Dr. Ben Swarts. Unless otherwise indicated, mid-log *M. smegmatis* or, where applicable, *B. subtilis*, *S. aureus*, *E. coli*, *L. lactis*, or *C. crescentus* were labeled with 2 mM alkDADA, 250 µM N-AlkTMM, 50 µM O-AlkTMM, or 5 µg/mL FM4-64FX for 15 min or 5 min in the case of *B. subtilis*. Unless otherwise indicated and where applicable, the bacteria were pre-incubated in the presence or absence of freshly prepared chemicals (antibiotics or benzyl alcohol) before being subjected to the probes (see *Supplementary files 1a,b*). Cells were pelleted by centrifugation, washed in PBST containing 0.01% BSA (PBSTB), and fixed for 10 min in 2% formaldehyde at room temperature. For alkDADA and O-AlkTMM, cells were further washed twice with PBSTB and subjected to CuAAC as described (*García-Heredia et al., 2018*; *Siegrist et al., 2013*). Unless otherwise specified, picolyl azide-Cy3 was used in *Figure 2D*; picolyl azide carboxyrhodamine 110 was used in *Figure 3B,D* and *Figure 2—figure supplement 3*, *Figure 3—figure supplements 4–6*; and 5-TAMRA picolyl azide was used in *Figure 3C*. Bacteria were then washed twice in PBSTB and once in PBST, and imaged (described below) or subjected to flow cytometry (BD DUAL LSRFortessa, UMass Amherst Flow Cytometry Core).

## Microscopy and image analysis

Bacteria were imaged on agar pads by either conventional fluorescence microscopy (Nikon Eclipse E600, Nikon Eclipse Ti, or Zeiss Axioscope A1 with 100× objectives) or structured illumination microscopy (Nikon SIM-E/A1R with SR Apo TIRF 100× objective, UMass Amherst Light Microscopy Core).

To obtain the fluorescence intensity plots, the subcellular distribution of fluorescence was quantitated from images obtained by conventional fluorescence microscopy. The images were processed using Fiji and Oufti (*Paintdakhi et al., 2016*; *Schindelin et al., 2012*) as described (*García-Heredia et al., 2018*). The signal was normalized to length and total fluorescence intensity of the cell. Cells were oriented such that the brighter pole is on the right hand of the graph. The intensity plots from *Figure 2A* were made from 42<n<56 cells; for *Figure 3B*, from 14<n<70 cells.

To quantify the amount of fractionated cellular material in *Figures 3* and *4*, images were processed in ImageJ, such that the cellular material corresponding to either the IMD or PM-CW fractions was measured. We then subtracted the signal from a constant-sized area of the gradient tubes that did not contain visible cellular material.

## Flow cytometry

Where appropriate, fixed bacterial samples were resuspended in PBS and subjected to flow cytometry analysis using FITC and Texas Red channels on a BD DUAL LSRFortessa instrument (UMass Amherst Flow Cytometry Core). Fifty thousand events per sample were gated on forward scatter vs. side scatter using our previously established values for intact bacterial cells.

## Membrane-bound peptidoglycan precursor analysis

Wild-type or MurJ-depleted *M. smegmatis* (*Gee et al., 2012*) were grown to mid-log phase and membrane fractions were isolated as above. Precursors were extracted from each membrane fraction similar to previous publications (*García-Heredia et al., 2018*; *Qiao et al., 2014*; *Qiao et al., 2017*). Briefly, glacial acetic acid was added to 500 µL of fractionated lysate to a final volume of 1%. The sample was then transferred into a vial containing 500 µL of chloroform and 1 mL of methanol and left at room temperature for 1–2 hr with occasional vortexing. The mixture was centrifuged at 21,000 x *g* for 10 min, and the supernatant was transferred into a vial containing 500 µL of 1% glacial acetic acid (in water) and 500 µL chloroform, and vortexed for 1 min. The mixture was separated by centrifugation (900 x *g* for 1 min at room temperature), and the organic phase was collected. Where applicable, the white interface was reextracted to recover additional organic material. The organic phase was dried under a nitrogen stream, and the dried lipids were resuspended in 12 µL of DMSO. D-Amino acid-containing lipid-linked peptidoglycan precursors were biotinylated by subjecting organic extracts to an in vitro PBP4-mediated exchange reaction with biotin-D-lysine (Chem-Impex International, Wood Dale, IL; reagent was deprotected first) as described (*Qiao et al., 2014*). The products were separated by SDS–PAGE on an 18%, polyacrylamide gels then transferred to a PVDF membrane, blotted with streptavidin-HRP (diluted 1:10,000; ThermoFisher Scientific, Waltham, MA), and visualized by ECL as above.

## Membrane glycolipid analysis

Sucrose density gradient fractions from wild-type *M. smegmatis* +/- 1 hr of 100 mM benzyl alcohol treatment or DivIVA-eGFP-ID with DivIVA depleted or not (*Meniche et al., 2014*) were subjected to lipid purification and analysis as previously described (*Morita et al., 2005*).

## Phylogenetic analysis

The phylogenetic tree was made in Adobe Illustrator (version 23.0.3) based on a phylogenetic tree generated with Mega (version 7.0.26; *Kumar et al., 2018*). Briefly, 16S rDNA sequences were obtained from NCBI (see *Supplementary file 1c*) and aligned using ClustalW. The phylogenetic tree was generated using the Timura–Ney model with Gamma distribution and Bootstrap method (C000 replications). The taxonomic information was verified with the Interagency Taxonomic Information System (available online https://www.itis.gov/).

## Acknowledgements

We are grateful to Drs. James Chambers and Amy Burnside for microscopy and flow cytometry guidance, Drs. Eric Strieter and Jiaan Liu and Ms. Sylvia Rivera and Katherine Chacón-Vargas for technical assistance. We also thank Drs. Hesper Rego, Karen Kieser and Eric Rubin for *ponA1-mRFP M. smegmatis,* Dr. Christopher Sassetti for the MurJ and MurG depletion strains, and Dr. Benjamin Swarts for O-AlkTMM and N-AlkTMM. Research was supported by funds from the National Institutes of Health (NIH) under awards R21 AI144748 (YSM and MSS), U01 CA221230 and NIH DP2 AI138238 (MSS) and R03 AI140259-01 (YSM), and R01 AI097191 (JJ and TAG). AG-H was supported by an Honors Fellowship from Universidad Autónoma de Nuevo León. JP is a recipient of the Science Without Borders Fellowship from CAPES-Brazil (0328-13-8). TK was supported by a postdoctoral fellowship from Uehara Memorial Foundation.

## Additional information

### Funding

| Funder | Grant reference number | Author |
|---|---|---|
| National Institutes of Health | R21 AI144748 | Yasu S Morita<br>M Sloan Siegrist |
| National Institutes of Health | U01 CA221230 | M Sloan Siegrist |
| National Institutes of Health | DP2 AI138238 | M Sloan Siegrist |
| National Institutes of Health | R03 AI140259-01 | Yasu S Morita |
| National Institutes of Health | R01 AI097191 | Todd A Gray |
| Universidad Autónoma de Nuevo León | Honors Fellowship | Alam García-Heredia |
| Uehara Memorial Foundation | Postdoctoral Fellowship | Takehiro Kado |
| Coordenação de Aperfeiçoamento de Pessoal de Nível Superior | Science Without Borders Fellowship (0328-13-8) | Julia Puffal |

The funders had no role in study design, data collection and interpretation, or the decision to submit the work for publication.

### Author contributions

Alam García-Heredia, Conceptualization, Resources, Data curation, Formal analysis, Supervision, Validation, Investigation, Visualization, Methodology, Writing - original draft, Writing - review and editing; Takehiro Kado, Julia Puffal, Resources, Data curation, Formal analysis, Validation, Investigation, Methodology, Writing - review and editing; Caralyn E Sein, Sarah H Osman, Data curation, Formal analysis, Validation, Investigation, Methodology, Writing - review and editing; Julius Judd, Resources, Validation, Methodology, Writing - review and editing; Todd A Gray, Resources, Validation, Visualization, Methodology, Writing - review and editing; Yasu S Morita, Conceptualization, Resources, Data curation, Formal analysis, Supervision, Funding acquisition, Validation, Investigation, Visualization, Methodology, Project administration, Writing - review and editing; M Sloan Siegrist, Conceptualization, Resources, Data curation, Formal analysis, Supervision, Funding acquisition, Validation, Visualization, Methodology, Writing - original draft, Project administration, Writing - review and editing

### Author ORCIDs

Alam García-Heredia ⬤ https://orcid.org/0000-0002-9573-4087
Julia Puffal ⬤ https://orcid.org/0000-0003-3066-5225
Julius Judd ⬤ https://orcid.org/0000-0002-4602-0205
Yasu S Morita ⬤ https://orcid.org/0000-0002-4514-9242
M Sloan Siegrist ⬤ https://orcid.org/0000-0002-8232-3246

### Decision letter and Author response

Decision letter https://doi.org/10.7554/eLife.60263.sa1
Author response https://doi.org/10.7554/eLife.60263.sa2

## Additional files

### Supplementary files

• Supplementary file 1. Supplementary information. (**a**) Incubation conditions. The table describes the incubation conditions employed in this study, including the concentrations and exposure times to a reagent. (**b**) Metabolic labeling conditions The table highlights the incubation conditions in which metabolic labeling was performed. (**c**) NCBI Accession numbers from 16S rDNA. This table

provides the NCBI accession numbers of the rDNA sequences used to create the phylogenetic tree in *Figure 3D*.

## Data availability

All of the source data used in this study is deposited in Open Science Framework (https://osf.io.10.17605/OSF.IO/FM794) and available for all public.

The following dataset was generated:

| Author(s) | Year | Dataset title | Dataset URL | Database and Identifier |
|---|---|---|---|---|
| Siegrist MS, Garcia-Heredia A | 2020 | The mycobacterial cell wall partitions the plasma membrane to organize its own synthesis | https://doi.org/10.17605/OSF.IO/FM794 | Open Science Framework, 10.17605/OSF.IO/FM794 |

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
