## [Decision Letter]

**Acceptance summary:**

This work should appeal to scientists interested in bacterial cell biology, especially those working on membrane organization and cell wall. The data convincingly shows that, in mycobacteria, the plasma membrane organizes into distinct functional domains that are important for the synthesis of the cell wall, whose physical structure feeds back to maintain membrane organization. The authors also present data suggesting that this connection between membrane domains and cell wall structure is conserved in diverse rod-shaped bacteria.

**Decision letter after peer review:**

Thank you for submitting your article "The mycobacterial cell wall partitions the plasma membrane to organize its own synthesis" for consideration by *eLife*. Your article has been reviewed by three peer reviewers, one of whom is a member of our Board of Reviewing Editors, and the evaluation has been overseen by Gisela Storz as the Senior Editor. The reviewers have opted to remain anonymous.

The reviewers have discussed the reviews with one another and the Reviewing Editor has drafted this decision to help you prepare a revised submission.

Summary:

In the work by García-Heredia et al., the authors took advantage of the fact that membrane domains are easily identified through microscopy and separated biochemically in Mycobacterium to determine whether the cell wall precursor lipid II synthesis is spatially separated from cell wall syntheses via partitions in the plasma membrane. The authors demonstrate that Lipid II is synthesized in the intracellular membrane domains (IMD), next to the polar, conventional plasma membrane associated with cell wall (PM-CW), in which lipid II is flipped and incorporated in the peptidoglycan cell wall. The authors show that this spatial separation of membrane domains is altered, and cell wall synthesizes reduced, when the membrane fluidity is perturbed or when the cell wall-organizing protein DivIVA is depleted. The authors further show that in spheroplast where the cell wall is digested away, the segregation of membrane domains is impaired. The authors propose that this type of membrane segregation is specifically important for directing cell wall synthesis in rod-shaped bacteria. While it remains to be determined whether this experimental asset and reported findings apply to other polar-growing bacteria remains to be determined, this system is interesting and medically relevant in and of itself, and contributes to our poor understanding of the existence and functional relevance of membrane domains in bacteria. However, the manuscript needs to be significantly strengthened to support their conclusions and improved for clarity.

Essential revisions:

1) Functionality of fusion constructs. In Figure 1C, it looks like MurG-Dendra2 is mostly in the cytoplasm with some in the IMD fraction. Figure 2A shows PonA1-mRFP as diffusive and cytoplasmic rather than bound on the membrane such as what is found in Figure 2—figure supplement 1A. Additionally, the authors cite Kieser et al. (PMID 26114871) for the functionality test of PonA1-mRFP, which further cites Hett et al. (PMID 20686708). Neither study had proper functionality tests besides growth curves. These inconsistencies raise concerns about the validity of these constructs and therefore the quality of the authors' conclusions. Please perform at least some of the following to confirm the functionality of these fusion proteins in addition to the depletion growth curves:

a) Western blot to show that in the depletion conditions there is no cleavage of the fluorescent tag of RFP or Dendra2 from MurG or PonA1. Cleavages of the tags would render untagged MurG or PonA to complement functionality and diffusive fluorescence.

b) Cell phenotype metrics such as length and morphology in addition to growth curves to compare with that of the wild type.

c) Show that the pole localization pattern of MurG-Dendra2 is not caused by the Dendra2 tag by swapping the tag with other true monomeric fluorescent proteins, and/or by assay untagged MurG in different cell fractions.

2) Quantifications of results. Most results were shown as representative images. It is unclear how large the sample size is and whether the differences are statistically significant. Some examples are listed below:

a) Figure 1C: the top gel showed that there are more MurG-Dendra2 in IMD compared to PM-CW. The apparent band density does not automatically translate into the enrichment of MurG-Dendra2 in IMD, especially given that MurG-Dendra2 is also present in the PM-CW fraction as well. Please quantify the amount of MurG in each fraction, normalize to the relative level of these two membrane fractions, and calculate the relative enrichment fold in IMD. The same applies to PonA1 and other IMD/PM-CW markers/proteins used in the study, as long as the protein is present in both fractions.

b) Figure 2A, Figure 2—figure supplement 1, Figure 3—figure supplement 4 and others: can colocalization or non-colocalization be quantified and averaged across many cells? In some cases, such as mCherry-GlfT2 and MurG-Dendra2 it appears to be clear, but in some other cases such as PonA1-mRFP and PimE-GFP it is not. The intensity-based line-scan can be used as a first level of colocalization analysis but it needs be done and averaged for all cells, not just on a selected few.

c) Reduced PG synthesis in Figure 2—figure supplement 2, Figure 3B, Figure 3C: can this be shown with quantification as that in Figure 3E using FACS (or just quantify the average fluorescence level of individual cells in microscopic images)? The authors argued that the effect of BA and DivIVA depletion was not additive so some numbers other than just the gel picture would help make the point.

d) Sucrose density gradient images (Figure 3A, 4A) are impossible to discern for untrained eyes. Quantification is necessary to show the differences.

3) Conclusions:

a) In BA- and Dibucaine-treated cells, the authors observed mis localized MurG-Dendra2 and PonA1 and reduced PG synthesis. It is reasonable to expect that membrane protein localizations are perturbed with these membrane-inserting/iron channel binding drugs, but it is difficult to draw the conclusion that reduced PG synthesis is also due to disturbed membrane architecture. Many other alternatives exist. For example, a drug could prevent protein-protein interactions to cause disruption in other enzymes' activities or disable membrane potentials to cause malfunction of membrane proteins. It is known that PG synthesis (and maybe even flipping activity) occur in large protein complexes. Similarly, the difference in the effect of BA-treated cells in different species (Figure 3E) could also be due to reasons other than the membrane domain architecture. These alternatives need to be discussed. Additionally, the working mechanisms of BA and Dibucaine are distinct, and the authors indeed showed that the time points for each treatment are significantly different (Figure 3B). However, the authors treated them the same way as to perturb membrane organization. Further explanations and justifications are needed.

b) The data set in Figure 3E is encouraging but needs additional support. The authors should further support their proposal that rod-shaped bacteria may rely on membrane fluidity and domain segregation for proper cell wall biogenesis by referencing some previous studies. Specifically, the authors should incorporate into their manuscript the recently published work by Zielinska et al., 2020. This new publication shows that membrane fluidity, which is controlled by flotillins, regulates cell wall synthesis and MreB movement. Along the same lines, the authors should also include earlier references about lipid II segregation to membrane domains and the fact that disrupting cell wall synthesis affects membrane domains (many references are listed in Zielinska et al.).

c) DivIVA creates/maintains IMD: Depleting DivIVA causes a large change in protein distributions between IMD and PM-CW, but such change could be induced by effects other than the disruption of IMD and PM-CW. For example, the changed cell shape could cause membrane curvature-sensing proteins to mis localize, or the space between the plasma membrane and cell wall being different to alter PG enzyme's activation. Alternatives should be included and discussed.

d) To truly demonstrate that IMD and PM-CW organization is important for cell wall synthesis, the authors should at least try some experiments in which they switch which domain MurG and DivIVA are by using different membrane tags to target them to the other compartment and see if PG synthesis is altered. Or, are the lipid compositions of the two compartments known? Can the authors artificially increase one over the other and see if the corresponding protein (MurG or DivIVA) goes with it? Most results presented in the study are consistent with the spatially segregated PG synthesis, but far from demonstrating that such spatial segregation is critical for PG synthesis. Conclusions could be rewritten to show that results suggest but not demonstrate.

e) The spheroplast experiment at the end: It suggests, but does not demonstrate that cell wall is required for membrane compartmentalization. For example, in spheroplasts, the whole gene expression profile also changed. One cannot say that it demonstrates that cell wall is required for gene expression.

4) Data representation and clarification:

a) The authors should clearly state at the beginning where IMD and PM-CW are in cells. From Figure 1 and 2 it appears that IMD is at cell pole and PM-CW is along the cell body, which could be misleading. The model in 4E is also not explained in the main text at all.

b) Figure indexing: It is difficult to go back and forth between different figures with the indexing not following the order of description. For example: Figure 2—figure supplement 1 is referred in Figure 1; Figure 1—figure supplement 2C referred in Figure 2C, Figure 1—figure supplement 2D referred in Figure 3A; Figure 4B before Figure 3C.

---

## [Author Response]

Essential revisions:1) Functionality of fusion constructs. In Figure 1C, it looks like MurG-Dendra2 is mostly in the cytoplasm with some in the IMD fraction. Figure 2A shows PonA1-mRFP as diffusive and cytoplasmic rather than bound on the membrane such as what is found in Figure 2—figure supplement 1A. Additionally, the authors cite Kieser et al. (PMID 26114871) for the functionality test of PonA1-mRFP, which further cites Hett et al. (PMID 20686708). Neither study had proper functionality tests besides growth curves. These inconsistencies raise concerns about the validity of these constructs and therefore the quality of the authors' conclusions. Please perform at least some of the following to confirm the functionality of these fusion proteins in addition to the depletion growth curves:a) Western blot to show that in the depletion conditions there is no cleavage of the fluorescent tag of RFP or Dendra2 from MurG or PonA1. Cleavages of the tags would render untagged MurG or PonA to complement functionality and diffusive fluorescence.

*i) MurG-Dendra2*. Using our in-gel fluorescence assay, we do not find a significant difference between the levels of MurG-Dendra2 +/- endogenous MurG, nor do we detect different-sized bands indicative of cleavage. We have included these results in Figure 1—figure supplement 1.

*ii) PonA1-mRFP.* To test the functionality of the *ponA1-mRFP* construct we have now performed an allele swap (Figure 2—figure supplement 1) and shown that the fusion protein is active (labeled by Bocillin-FL) in the PM-CW (Figure 2—figure supplement 2), similar to endogenous protein (Figure 1C).

We do find that there are different-sized bands by anti-RFP immunoblot. We have included these data in Figure 2—figure supplement 1, and noted them in the text. Unfortunately, to our knowledge there are no other fluorescent proteins that are suitable for tagging secreted proteins in mycobacteria (based on the literature, conversations with other researchers in the field, and our own experience). Because we cannot currently exclude the possibility that the breakdown products are fluorescent we have tempered our claim.

b) Cell phenotype metrics such as length and morphology in addition to growth curves to compare with that of the wild type.

We do not find significant changes in cell length for strains expressing the fluorescent fusion proteins (Figure 1—figure supplement 2) nor do we observe obvious differences in cell morphology.

c) Show that the pole localization pattern of MurG-Dendra2 is not caused by the Dendra2 tag by swapping the tag with other true monomeric fluorescent proteins, and/or by assay untagged MurG in different cell fractions.

We previously demonstrated similar localization patterns for other MurG fluorescent fusions (Meniche et al., 2014): MurG-GFP, under its own promoter and in native chromosomal context, and MurG-RFP, expressed episomally under a non-native promoter. Polar enrichment of MurG-RFP was spatially coincident with that of GlfT2-GFP (GlfT2 is an IMD marker, see Hayashi et al., 2016), a finding we were able to recapitulate in this study using MurG-Dendra2 and mCherry-GlfT2 (Figure 1—figure supplement 5, Figure 2, Figure 3—figure supplement 7).

While the above-referenced MurG fusions are not truly monomeric, we note that native MurG is enriched in the IMD proteome (Hayashi et al., 2016). Likewise we see that membrane-bound MurG-Dendra2 preferentially associates with the IMD (Figure 1C and Figure 1—figure supplement 4).

2) Quantifications of results. Most results were shown as representative images. It is unclear how large the sample size is and whether the differences are statistically significant. Some examples are listed below:a) Figure 1C: the top gel showed that there are more MurG-Dendra2 in IMD compared to PM-CW. The apparent band density does not automatically translate into the enrichment of MurG-Dendra2 in IMD, especially given that MurG-Dendra2 is also present in the PM-CW fraction as well. Please quantify the amount of MurG in each fraction, normalize to the relative level of these two membrane fractions, and calculate the relative enrichment fold in IMD. The same applies to PonA1 and other IMD/PM-CW markers/proteins used in the study, as long as the protein is present in both fractions.

We have quantified the signal from MurG-Dendra2 and Bocillin-FL and included the data as a new supplementary figure (Figure 1—figure supplement 4).

b) Figure 2A, Figure 2—figure supplement 1, Figure 3—figure supplement 4 and others: can colocalization or non-colocalization be quantified and averaged across many cells? In some cases, such as mCherry-GlfT2 and MurG-Dendra2 it appears to be clear, but in some other cases such as PonA1-mRFP and PimE-GFP it is not. The intensity-based line-scan can be used as a first level of colocalization analysis but it needs be done and averaged for all cells, not just on a selected few.

We have calculated Pearson´s coefficients for Figure 1—figure supplement 5, Figure 2A, Figure 2—figure supplement 2C, and Figure 3—figure supplement 7.

c) Reduced PG synthesis in Figure 2—figure supplement 2, Figure 3B, Figure 3C: can this be shown with quantification as that in Figure 3E using FACS (or just quantify the average fluorescence level of individual cells in microscopic images)? The authors argued that the effect of BA and DivIVA depletion was not additive so some numbers other than just the gel picture would help make the point.

We have now included quantitation in Figure 2—figure supplement 3B (right, new flow cytometry data), Figure 3C (below the immunoblot and right, new flow cytometry data), and Figure 3—figure supplement 5 (new flow cytometry data to accompany Figure 3B).

d) Sucrose density gradient images (Figure 3A, 4A) are impossible to discern for untrained eyes. Quantification is necessary to show the differences.

We have quantified the amount of cellular material present in the tubes by densitometry, now Figure 3—figure supplement 1.

3) Conclusions:a) In BA- and Dibucaine-treated cells, the authors observed mis localized MurG-Dendra2 and PonA1 and reduced PG synthesis. It is reasonable to expect that membrane protein localizations are perturbed with these membrane-inserting/iron channel binding drugs, but it is difficult to draw the conclusion that reduced PG synthesis is also due to disturbed membrane architecture. Many other alternatives exist. For example, a drug could prevent protein-protein interactions to cause disruption in other enzymes' activities or disable membrane potentials to cause malfunction of membrane proteins. It is known that PG synthesis (and maybe even flipping activity) occur in large protein complexes. Similarly, the difference in the effect of BA-treated cells in different species (Figure 3E) could also be due to reasons other than the membrane domain architecture. These alternatives need to be discussed. Additionally, the working mechanisms of BA and Dibucaine are distinct, and the authors indeed showed that the time points for each treatment are significantly different (Figure 3B). However, the authors treated them the same way as to perturb membrane organization. Further explanations and justifications are needed.

We agree that we cannot rule out pleiotropic effects of benzyl alcohol and dibucaine on, for example, membrane potential, membrane proteins or global gene expression. We now discuss this important caveat in the text as well as the in vitro mechanisms of action for benzyl alcohol and dibucaine in the text. We are currently investigating the in vivo mechanisms of these compounds and their effects on *M. smegmatis* in much greater depth via Tn-Seq and other experiments.

b) The data set in Figure 3E is encouraging but needs additional support. The authors should further support their proposal that rod-shaped bacteria may rely on membrane fluidity and domain segregation for proper cell wall biogenesis by referencing some previous studies. Specifically, the authors should incorporate into their manuscript the recently published work by Zielinska et al., 2020. This new publication shows that membrane fluidity, which is controlled by flotillins, regulates cell wall synthesis and MreB movement. Along the same lines, the authors should also include earlier references about lipid II segregation to membrane domains and the fact that disrupting cell wall synthesis affects membrane domains (many references are listed in Zielinska et al.).

We agree that these references are critical to include in the manuscript. Because of the complexity of the system, and our focus on mycobacteria, we have tried to organize the manuscript in such a way that it systematically addresses each piece of the model in Figure 4E (in the order of arrow 2 -> arrow 1-> arrow 3). In the context of the present manuscript, we view the data in Figure 3E primarily as introducing the idea that there may be a conserved role for the cytoskeleton in membrane partitioning and for setting up the hypothesis that DivIVA fulfills this role in mycobacteria *i.e.* arrow 1 in the Figure 4E model. Rather than put all of the above references with Figure 3E we have interwoven them in the manuscript according to the part of the model and set of *M. smegmatis* experiments that they inspire:

conserved role of cytoskeleton in membrane partitioning in non-mycobacterial species i.e. arrow 1 (Strahl et al., 2014; Oswald et al., 2016)

testing whether there is feedback from the membrane to DivIVA i.e. reverse of arrow 1 (Zielinska et al., 2020)

discussion of whether localized cell wall synthesis is a cause (arrow 3) or consequence (arrow 2) of membrane organization (Ganchev et al., 2006; Jia et al., 2011; Valtersson et al., 1985; Dominguez-Escobar et al., 2011; Garner et al., 2011; van Teeffelen et al., 2011; Schirner et al., 2015; Muchova et al., 2011)

c) DivIVA creates/maintains IMD: Depleting DivIVA causes a large change in protein distributions between IMD and PM-CW, but such change could be induced by effects other than the disruption of IMD and PM-CW. For example, the changed cell shape could cause membrane curvature-sensing proteins to mis localize, or the space between the plasma membrane and cell wall being different to alter PG enzyme's activation. Alternatives should be included and discussed.

We now discuss these possibilities.

d) To truly demonstrate that IMD and PM-CW organization is important for cell wall synthesis, the authors should at least try some experiments in which they switch which domain MurG and DivIVA are by using different membrane tags to target them to the other compartment and see if PG synthesis is altered. Or, are the lipid compositions of the two compartments known? Can the authors artificially increase one over the other and see if the corresponding protein (MurG or DivIVA) goes with it? Most results presented in the study are consistent with the spatially segregated PG synthesis, but far from demonstrating that such spatial segregation is critical for PG synthesis. Conclusions could be rewritten to show that results suggest but not demonstrate.

We agree; these experiments are in progress:

*i) Compartment switching*. While we do not yet know what makes a protein associate with a given membrane compartment in mycobacteria, we think we may have a lead. We have two fluorescent protein fusion constructs for a PM-CW-associated protein (determined in Hayashi et al., 2016). One is non-toxic, localizes around the perimeter of the cell, and, like the native protein, is present and active in the PM-CW. The other is toxic, localizes to the poles, and is present in the IMD. There is a point mutation in the toxic construct that we hypothesize explains the difference in localization. If true, we plan to test whether similar mutations can make other, normally-PM-CW proteins go to the IMD.

*ii) Tuning lipid composition.* The IMD lipidome is distinct from that of the PM-CW (Morita et al., 2005; Hayashi et al., 2016). We hypothesize that a subset of these lipids help establish and/or maintain the membrane domain and are testing this using both genetic and chemical perturbations.

Because both of these studies are in their infancy, for the current manuscript we have opted to temper our language as suggested.

e) The spheroplast experiment at the end: It suggests, but does not demonstrate that cell wall is required for membrane compartmentalization. For example, in spheroplasts, the whole gene expression profile also changed. One cannot say that it demonstrates that cell wall is required for gene expression.

We agree and have changed the text in the Title, Abstract and main document accordingly. We have also referenced a recent preprint (*bioRxiv* 060970), in which the authors suggest a similar role for the *B. subtilis* cell wall in segregating membrane domains.

4) Data representation and clarification:a) The authors should clearly state at the beginning where IMD and PM-CW are in cells. From Figure 1 and 2 it appears that IMD is at cell pole and PM-CW is along the cell body, which could be misleading. The model in 4E is also not explained in the main text at all.

We have clarified the locations of the IMD and PM-CW in the Introduction and our model in Figure 4E.

b) Figure indexing: It is difficult to go back and forth between different figures with the indexing not following the order of description. For example: Figure 2—figure supplement 1 is referred in Figure 1; Figure 1—figure supplement 2C referred in Figure 2C, Figure 1—figure supplement 2D referred in Figure 3A; Figure 4B before Figure 3C.

We have rearranged the order of the figures to follow the order of descriptions.